# Racial Disparities in Hepatitis B Birth Dose in the Washington Metropolitan Region, 2018–2020

**DOI:** 10.3390/vaccines10071121

**Published:** 2022-07-14

**Authors:** Hee-Soon Juon, Donna T. Sheler, Jane Pan, Daisy Le, Y. Tony Yang

**Affiliations:** 1Department of Medical Oncology, Thomas Jefferson University, 834 Chestnut Street Suite 314, Philadelphia, PA 19107, USA; 2DC Health Department, HIV/AIDS Hepatitis STD TB Administration, Washington, DC 20002, USA; donna.sheler@dc.gov; 3Hepatitis B Initiative of Washington DC (HBI-DC), Washington, DC 20006, USA; janepan@hbi-dc.org; 4Department of Policy, Populations, and Systems, School of Nursing, George Washington University, Washington, DC 20006, USA; daisyle@email.gwu.edu (D.L.); ytyang@email.gwu.edu (Y.T.Y.)

**Keywords:** hepatitis B birth dose, race, retrospective analysis, primary prevention

## Abstract

Hepatitis B vaccination protects newborns from contracting the hepatitis B virus that may lead to chronic infection, liver failure, or death. Trends and racial differences in the administration of the hepatitis B (HepB) birth dose in 2018–2020 were examined in the targeted region. A retrospective analysis of electronic birth dose vaccination data of newborns in 2018–2020 was performed. Birth data from six birthing facilities and home delivery records were obtained from the DC Health Department Vital Statistics Division. This data represented 40,269 newborns and included the mother’s race and ethnicity, health insurance type, birthing facility, and administration of the HepB birth dose. Descriptive analysis and multivariable logistic regression analysis were conducted. In addition, subgroup analysis by health insurance type was also conducted with a significant interaction of race/ethnicity and health insurance type. A total of 34,509 (85.7%) received the HepB birth dose within 12 h or before discharge from the facility. The rates of birth dose vaccination have seen an increase over the 3-year period (83.7% in 2018, 85.8% in 2018, 87.7% in 2020, *p* < 0.01). Multivariable logistic regression analysis revealed racial differences in HepB birth dose vaccination rates. Asian Americans had the highest rate of newborn vaccination consistently over the 3-year period. Conversely, African American infants were less likely to have the birth dose than non-Hispanic Whites (aOR = 0.77, 95% CI: 0.71–0.83). Our research indicates that further studies are needed to explore HepB birth dose hesitancy among African Americans.

## 1. Introduction

Universal hepatitis B (HepB) vaccination among newborns is critical to the public health effort to eliminate HepB infection in the United States (US) [1]. HepB vaccination protects newborns from contracting the hepatitis B virus (HBV) that may lead to chronic infection, liver failure, or death. In October 2016, the Advisory Committee on Immunization Practices (ACIP) recommended that all US-born infants who weigh at least 2000 g receive a dose of the HepB vaccine before 24 h of age [2]. The American Academy of Pediatrics endorses this recommendation in its policy statement [3]. The 2020 National Immunization Survey reports that 78.4% of newborns received their first HepB vaccine dose within 3 days of birth (2017–2018) [4]. Healthy People 2030 aims to increase HepB birth dose rates up to 90% [5]. The purpose of this study was to examine the trend in Hep B birth dose rates in the Washington metropolitan region for the years 2018–2020 and the racial/ethnic differences in the birth dose administration.

## 2. Methods

Data were obtained from the Vital Statistics Division of the District of Columbia Health Department (DC DOH). A retrospective analysis was performed using electronic birth dose vaccination entries of newborns in the 2018–2020 cohorts. Six birthing facility and home delivery records representing 40,269 newborns were collected (237 were excluded due to missing race/ethnicity data). Data variables included the mother’s race and ethnicity, health insurance type, birthing facility, and the HepB birth dose administration date, lot number, and manufacturer. Descriptive and logistic regression analyses were conducted. The DC DOH Institutional Review Board waived review because the study did not meet the criteria for human subject research.

## 3. Results

The HepB birth dose vaccination rates for the District of Columbia subsequently increased over the three-year period (83.7% in 2018, 85.8% in 2018, 87.7% in 2020, *p* < 0.01) (see Figure 1). Table 1 shows this trend in the HepB birth dose by race/ethnicity over the 3-year period: NHW, Hispanics, and Asians increased the rates of HepB birth dose, while African Americans decreased rates over the three-year period.

A total of 40,032 newborns made up the sample. Most were African Americans (42%) followed by Non-Hispanic Whites (NHW) (40%), Hispanics (11%), and Asians (6%). Birthing facilities included community hospitals (57%) and university-affiliated hospitals (42%). Less than one percent had a home or birthing center delivery. Two-thirds of mothers had private health insurance (see Table 2).

Table 2 shows the bivariate relationship of birth dose. Year, facility, and race/ethnicity were highly associated with having the birth dose (*p* < 0.001), while health insurance type was not associated with it. Overall, there were racial/ethnic differences in the HepB birth dose rates. Asian newborns (89.7%) had the highest vaccination rate followed by Hispanics (89.3%), then NHW (86.2%). African Americans (83.9%) had the lowest newborn vaccination rate.

In addition, race/ethnicity and health insurance type were highly associated (*p* < 0.001); about two-thirds of African Americans (62.9%) had Medicaid followed by Hispanics (38.5%), Asians (6.1%), and NHW (3.7%) (table not shown). Since the interaction between race/ethnicity and health insurance type on HepB birth dose was statistically significant (see Figure 2), subgroup analyses by health insurance type were conducted.

Table 3 shows multivariable logistic regression analysis and subgroup analysis by insurance type. Year, facility, race/ethnicity, and health insurance type were associated with the outcome. There were racial/ethnic differences in the HepB birth dose: African American newborns were less likely to receive the HepB vaccine within 24 h of birth than NHW (aOR = 0.77, 95% CI: 0.71–0.83). Furthermore, Asians (aOR = 1.38, 95% CI: 1.20–1.59) and Hispanics (aOR = 1.22, 95% CI: 1.09–1.37) were more likely to receive the birth dose than NHW. In subgroup analysis, race/ethnicity had different effects on birth dose depending on the health insurance type. Among those with private insurance, African Americans (aOR = 0.72, 95% CI = 0.66–0.78) were less likely to have birth dose than NHW, while Asians (aOR = 1.41, 95% CI = 1.22–1.64) were more likely to have birth dose than NHW. Among those with Medicaid and self-pay/others, Hispanics were more likely to have birth doses than NHW (aOR = 1.74 for Medicaid; aOR = 2.15 for self-pay/others).

## 4. Discussion

This study indicates that the overall HepB birth dose vaccination rate has increased among facilities in the Washington metropolitan region from 2018 to 2020. DC DOH policy and program activities include verifying the documentation of newborn HepB vaccination in hospital birth records and entering birth dose data into the D.C. Immunization Information System (DOCIIS) registry. Timely identification of newborns at risk of Hep B infection and intervention during perinatal care, such as HBV load monitoring, are important steps for reducing the transmission of hepatitis B. By following the Center for Disease Control’s state perinatal hepatitis B prevention program, the District of Columbia (87.7%) is close to reaching the Healthy People 2030 objectives (90%).

In our subgroup analysis, the HepB birth dose rates increased for newborns across all racial/ethnic populations except for African Americans. The rate for this subgroup did not change over the 3-year period. After adjusting for the year, birthing facility, and health insurance, African Americans had the lowest rates of receiving the HepB birth dose. In recognition of these findings, exploration of birth dose hesitancy among African Americans is needed.

The results of our interaction analyses between race/ethnicity and health insurance type are interesting. Although health insurance type was not associated with HepB birth dose in our bivariate analysis, there was a main effect of insurance type on HepB birth dose after adjusting all the variables. It may be explained by two factors: (1) there was a larger representation of African Americans in our study sample (42% from the Washington metropolitan area vs. 12.4% of the total U.S. population in 2020) [6]. (2) African Americans had the highest proportion of Medicaid recipients in the study. Further research is needed to explore this effect in other geographic areas.

There are some limitations to our study. First, our registry did not have data on mother-related characteristics including age, education, and hepatitis B antigen status, which may be important factors associated with HepB birth dose. Additional studies examining the specific impact of these mother-related variables on HepB birth dose are recommended. Second, our findings are based on a concentrated population within the Washington metropolitan region and may have limited generalizability to the US population in other geographic settings.

In conclusion, worldwide, perinatal transmission occurring at birth from infected mothers to their newborns accounts for the majority of hepatitis B virus (HBV) transmissions [7]. In the US, Asian Americans make up 50% of the hepatitis infection burden, despite making up only 6% of the population. Many affected Asian Americans contracted the disease through vertical transmission [8]. For these reasons, giving the birth dose of the HepB vaccine is particularly valuable for Asian Americans and may be a plausible reason why in this study Asians had consistently higher rates of birth dose administration. Efforts need to continue to promote the timely administration of the HepB birth dose and administration of hepatitis B immune globulin (HBIG) where indicated to prevent vertical transmission (particularly in high-HBV-prevalent populations from East Asia and Sub-Saharan Africa). Our study recommends that a policy for universal vaccination of all newborns within 24 h of birth should be promoted amongst birthing facilities to prevent unrecognized perinatal transmission. Moreover, further research and effort are warranted related to vaccine hesitancy.

## Figures and Tables

**Figure 1 vaccines-10-01121-f001:**
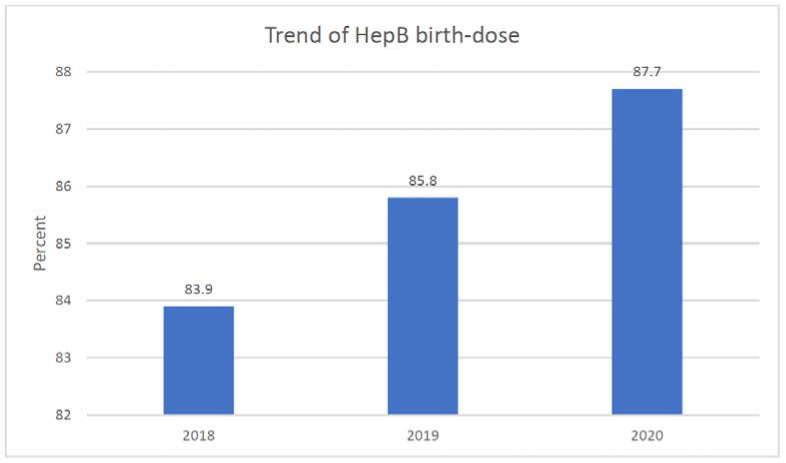
Trend of HBV birth dose rates in Washington metropolitan region, 2018–2020. Note *p* < 0.001.

**Figure 2 vaccines-10-01121-f002:**
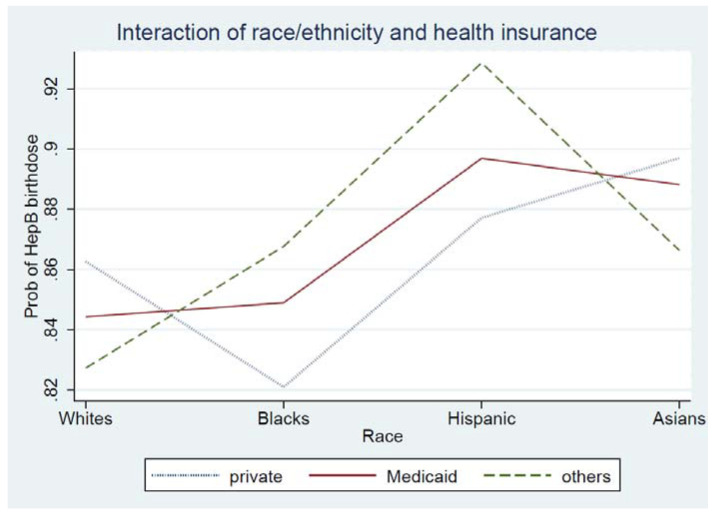
Interaction of race/ethnicity and health insurance type on HepB birth dose. Note *p* < 0.001.

**Table 1 vaccines-10-01121-t001:** HepB birth dose vaccination rates by race/ethnicity, 2018–2020.

		2018 %Having HepB Birth Dose	2019 %Having HepB Birth Dose	2020 %Having HepB Birth Dose
Mother’s Race	White	4522 (82.1)	4721 (86.1)	4798 (90.5)
	Black	5207 (83.6)	4690 (84.4)	4245 (83.7)
	Hispanics	1579 (88.7)	1265 (88.7)	1134 (90.9)
	Asian	768 (87.4)	765 (90.4)	637 (91.7)

**Table 2 vaccines-10-01121-t002:** Background characteristics and bivariate analyses of HepB birth dose, Washington metropolitan hospitals (*n* = 40,269), 2018–2020.

		*n* (%)	% Having Birth Dose
Year	2018	14,464 (35.9%)	83.9% *
	2019	13,399 (33.3%)	85.8%
	2020	12,406 (30.8%)	87.7%
Facilities	Academic affiliated Hospitals	17,028 (42.3%)	85.7% *
	Community Hospitals	22,999 (57.1%)	86.3%
	Others (Home Address)	225 (0.6%)	20.4%
Mother’s Race	Non-Hispanic Whites	16,299 (40.7%)	86.2% *
	African Americans	16,858 (42.1%)	83.9%
	Hispanics	4455 (11.1%)	89.3%
	Asians	2420 (6.1%)	89.7%
Health Insurance Type	Private	25,570 (64.1%)	85.9%
	Medicaid	12,977 (32.6%)	85.5%
	Self-pay/Others	1289 (3.3%)	84.7%

Note. * *p* < 0.01.

**Table 3 vaccines-10-01121-t003:** Multivariable logistic regression of HepB birth dose: by health insurance type.

		Total (*n* = 40,269)	Private (*n* = 25,434)	Medicare (*n* = 12,899)	Self-Pay/Others (*n* = 1305)
		Adjusted OR (95% CI)	Adjusted OR (95% CI)	Adjusted OR (95% CI)	Adjusted O (95% CI)
Year	2018	1	1	1	1
	2019	1.21(1.13–1.30) *	1.36(1.25–1.48) *	1.00(0.88–1.12)	1.12(0.61–2.05)
	2020	1.44(1.34–1.55) *	1.82(1.67–2.00) *	1.01(0.90–1.14)	0.88(0.57–1.35)
Facilities	Academic affiliated Hospitals	1	1	1	1
	Community Hospitals	1.05(0.98–1.11)	1.32(1.23–1.42) *	0.71(0.65–0.79) *	0.72(0.50–1.03)
	Others (Home Address)	0.04(0.03–0.05) *	0.02(0.01–0.03) *	0.25(0.14–0.46) *	0.01(0.00–0.02) *
Mother’s Race	Non-Hispanic Whites	1	1	1	1
	African Americans	0.77(0.71–0.83) *	0.72(0.66–0.78) *	1.05(0.83–1.31)	1.18(0.73–1.91)
	Hispanics	1.22(1.09–1.37) *	1.15(0.99–1.32)	1.74(1.32–2.29) *	2.15(1.24–3.71) *
	Asians	1.38 (1.20–1.59) *	1.41(1.22–1.64) *	1.64(0.93–2.89)	1.19(0.51–2.76)
Health Insurance Type	Private	1	N/A	N/A	N/A
	Medicaid	1.17(1.08–1.26) *			
	Self-pay/Others	1.30(1.09–1.55) *			

Note. * *p* < 0.05.

## Data Availability

The data presented in this study are available upon request from the corresponding author. The data are not publicly available due to restrictions of the DC Department of Health.

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
