# Peer review of "Racial Disparities in Hepatitis B Birth Dose in the Washington Metropolitan Region, 2018–2020"

_vaccines, 2022, doi:10.3390/vaccines10071121_

Round 1

Reviewer 1 Report

The topic of the work is interesting. I appreciate the scope of the cohort and the large number of patients enrolled. On the other hand, the amount of data obtained is small and relevant results are minimal. To improve the attractiveness of the article, it would be useful to add additional data, if available. In this regard, I have some questions or suggestions for the authors:

1.    Lines 67-69 state: „In our subgroup analysis, the HepB birth dose rates increased for newborns across all racial/ethnic populations except for African Americans: its rate among African Americans did not change over the 3-year period“. Data on the subgroup analysis by years according to different race or ethnicity are not given either in the results section or in the table.  Can you provide these values as well?

2.    Is it possible to state what was the proportion of HBsAg positive mothers in total and in all racial/ethnic subgroups? The prevalence of HBV infection may vary significantly by race or ethnicity.

3.    Are data on HBeAg status and HBV DNA values from mothers available?

4.    Did newborns who were not vaccinated within 24 hours receive the vaccine later? In the case of children of HBsAg negative mothers, such a delay would not be a problem. Or were these children not vaccinated at all?

5.    If patient's race or ethnicity is an independent risk factor (not related, for example, to the health insurance status) for receiving HepB birth dose, how do you explain this fact? Do you have at least some theory to explain these findings?

Author Response

RESPONSE TO REVIEWER COMMENTS

Thank you to the reviewers for their constructive comments. We have provided point-by-point responses to the reviewers’ comments and suggestions with our specific changes. We hope that these changes have improved our manuscript so that it is now acceptable to be published in the Vaccines.

Reviewer 1

The topic of the work is interesting. I appreciate the scope of the cohort and the large number of patients enrolled. On the other hand, the amount of data obtained is small and relevant results are minimal. To improve the attractiveness of the article, it would be useful to add additional data, if available. In this regard, I have some questions or suggestions for the authors:

  1. Lines 67-69 state: „In our subgroup analysis, the HepB birth dose rates increased for newborns across all racial/ethnic populations except for African Americans: its rate among African Americans did not change over the 3-year period“. Data on the subgroup analysis by years according to different race or ethnicity are not given either in the results section or in the table. Can you provide these values as well?

Response: We added Table 1 to show race/ethnicity and birth dose rates by year. We also provided bivariate relationships between factors and HepB birth dose in Table 2.

  1. Is it possible to state what was the proportion of HBsAg positive mothers in total and in all racial/ethnic subgroups? The prevalence of HBV infection may vary significantly by race or ethnicity.

Response: We have the number of HBsAg positive mothers and their racial/ethnic groups for each year since 2016. But they are all DC residents, and our sample population are residents of multiple states including DC, Maryland, and Virginia. It would not be an accurate comparison.

  1. Are data on HBeAg status and HBV DNA values from mothers available?

Response: While we do believe that it could be possible for us to request HBV DNA results on most of the cases in this study (at least for those since 2017), the process involved would be time-intensive/laborious (involving approvals on multiple levels). We do not believe that providing the additional data on HBV DNA results would add value to this particular paper as it deviates from our original study objective and intent of submitting our findings as a brief report.

  1. Did newborns who were not vaccinated within 24 hours receive the vaccine later? In the case of children of HBsAg negative mothers, such a delay would not be a problem. Or were these children not vaccinated at all?

Response: Unfortunately, due to registry and data limitations, the following information (on infants from 2018-2020) is not available: those who did not get the birth dose vaccine; whether they got the vaccine subsequently; how long after; and whether they remained unvaccinated.

  1. If patient's race or ethnicity is an independent risk factor (not related, for example, to the health insurance status) for receiving HepB birth dose, how do you explain this fact? Do you have at least some theory to explain these findings?

Response: We appreciate the reviewer’s comment on the independent effect of race/ethnicity and health insurance. First, we examined the association of race and health insurance type which was statistically significant. African Americans had higher rates of having Medicaid than any other racial/ethnic groups. Then, we checked the interaction of race and health insurance type on HepB birth dose which was significant (see Figure 2). Thus, we conducted subgroup analysis by health insurance type in Table 3. We also included discussion on the results of interaction in consideration of overrepresentation of African Americans and high proportion of Medicaid recipients among African Americans in the Washington metropolitan region.

Reviewer 2 Report

I think this information is necessary to further raise the HepB birth dose coverage in the future.

Please address the following points,

P4, L74 (DISCUSSION)

[For this reason, giving the birth dose of HepB vaccine is of particular value and may be a plausible reason why Asian s have consistently higher rates of birth dose administration in this study.]

I think the explanation why Asian races have a high interest in vaccination is needed. For instance, “the prevalence of hepatitis B among general population in the western pacific region is high”.

Figure 1

There is no explanation for the orange bar graph. Please remove if unnecessary.

Could you please check the grammar of these sentences?

P1, L31: The purpose of this study to examine the trend of birth dose rates in 2018-2020 and racial/ethnic differences of birth dose in the Washington metropolitan region.

P2, L77: The importance of primary prevention from through vaccination in infancy to prevent long-term adverse outcome of HBV infection.

Author Response

[1] Regarding Reviewer 2's comment on P4, L74: We have now provided some context as to potential reasons why Asians may value the HepB vaccine.

[2] We have updated Figure 1.

[3] We have revised the sentence on P1, L31 as such: The purpose of this study is to examine the trend, and racial/ethnic differences, of birth dose rates in the Washington metropolitan region from 2018-2020.

[4] We have revised the sentence on P2, L77 as such: The importance of Primary prevention from through vaccination in infancy is critical to preventing long-term adverse outcomes from HBV infection.